# Enhancing Maize Yield and Quality with Metal-Based Nanoparticles without Translocation Risks: A Brief Field Study

**DOI:** 10.3390/plants13141936

**Published:** 2024-07-14

**Authors:** Dávid Ernst, Marek Kolenčík, Martin Šebesta, Veronika Žitniak Čurná, Yu Qian, Viktor Straka, Ladislav Ducsay, Gabriela Kratošová, Ľuba Ďurišová, Ján Gažo, Juraj Baláži

**Affiliations:** 1Institute of Agronomic Sciences, Faculty of Agrobiology and Food Resources, Slovak University of Agriculture in Nitra, Tr. A. Hlinku 2, 949 76 Nitra, Slovakia; david.ernst@uniag.sk (D.E.); veronika.curna@uniag.sk (V.Ž.Č.); straka.viktor.spu@gmail.com (V.S.); ladislav.ducsay@uniag.sk (L.D.); 2Institute of Laboratory Research on Geomaterials, Faculty of Natural Sciences, Comenius University in Bratislava, Mlynská Dolina, Ilkovičova 6, 842 15 Bratislava, Slovakia; martin.sebesta@uniba.sk; 3School of Ecology and Environmental Science, Yunnan University, 2 Cuihubei Lu, Kunming 650091, China; qianyu@ynu.edu.cn; 4Nanotechnology Centre, Centre for Energy and Environmental Technologies, VŠB Technical University of Ostrava, 17. Listopadu 15/2172, 708 00 Ostrava, Czech Republic; gabriela.kratosova@vsb.cz; 5Institute of Plant and Environmental Sciences, Faculty of Agrobiology and Food Resources, Slovak University of Agriculture in Nitra, Tr. A. Hlinku 2, 949 76 Nitra, Slovakia; luba.durisova@uniag.sk (Ľ.Ď.); jan.gazo@uniag.sk (J.G.); 6Institute of Design and Engineering Technologies, Faculty of Engineering, Slovak University of Agriculture in Nitra, Tr. A. Hlinku 2, 949 76 Nitra, Slovakia; juraj.balazi@uniag.sk

**Keywords:** spray application, maize, yield, grain quality, nanofertilizers, nanoparticles, gold, biosilica, zinc oxide, titanium dioxide

## Abstract

Our previous studies have shown physiological and yield intensification of selected crops with the application of nanoparticles (NPs). However, the impact on the quantitative, qualitative, and yield parameters of maize (*Zea mays* L.) in field conditions remains highly debated. This study aimed to evaluate the effects of zinc oxide (ZnO-NPs), gold NPs anchored to meso-biosilica (Au-NP-bioSi), and titanium dioxide (TiO_2_-NPs) as biological stimulants under field conditions during the vegetation season of 2021 in the Central European region. The study assessed the effects on the number of plants, yield, yield components, and nutritional quality, including mineral nutrients, starch, and crude protein levels. The potential translocation of these chemically–physically stable NPs, which could pose a hazard, was also investigated. The results indicate that Au-NP-bioSi and ZnO-NPs-treatments were the most beneficial for yield and yield components at a statistically significant level. Mineral nutrient outcomes were varied, with the NP-free variant performing the best for phosphorus-levels, while Au-NP-bioSi and ZnO-NPs were optimal for crude protein. Starch content was comparable across the TiO_2_-NPs, Au-NP-bioSi, and control variants. Importantly, we observed no hazardous translocation of NPs or negative impacts on maize grain quality. This supports the hypothesis that NPs can serve as an effective tool for precise and sustainable agriculture.

## 1. Introduction

Nanotechnology is becoming integrated into agriculture across various domains, including the production and utilization of nanofertilizers (NFs) [1]. Some pioneering studies focus on applications to enhance crop production using plant stimulators, such as Si- or Ti-based materials [2] or “exotic” precious metals, including gold nanoparticles (Au-NPs) [3]. For NPs to fall within the category of NFs, they must have at least one of the three dimensions measuring between 1 and 100 nm [4], where NPs may exhibit different mobility, reactivity, or potential toxicity than their similar ionic form and bulk counterparts [5]. According to Liu and Lal [2], nanoparticles (NPs) are classified based on essentiality into various categories. ZnO-NPs fall under micronutrient NFs, while silicon-based porous materials with Au-NPs attached belong to another category, specifically the kind of nanomaterial (NM)-enhanced fertilizers. This category also includes TiO_2_-NPs, carbon-based nano-sized materials, and similar NMs that are characterized as directly lacking nutrients while often providing agronomically beneficial effects stimulating growth, plant health, and yield parameters; however, the mechanisms of action are partially unknown.

In this context, the NF application offers several effects, including environmental benefits, due to the significantly lower concentrations utilized in comparison to conventional agronomic fertilizers. Additionally, gradual nutrient release, especially in the case of spray application, provides a more balanced agronomic impact [5]. Also, they do not pose a reproductive risk or processes related to original plant species or the diversity of agrocenoses [6]. One hypothesis suggests that foliar application of NPs, besides supplementing the micronutrients, may directly intensify photosynthetic processes. For instance, the application of ZnO-NPs, Au-NP-bioSi, or TiO_2_-NPs to sunflower has been shown to enhance photosynthesis [7,8]. Similarly, a positive effect of TiO_2_-NPs on photosynthesis was observed in the case of spinach [9]. Additionally, this beneficial effect was also associated with higher agronomic yields, as observed in foxtail millet [10] or lentil [11].

The literature largely lacks information on influence of NFs on qualitative parameters concerning human consumption [12], such as nutritional value, which can be evaluated based on mineral nutrients profile, starch, and crude protein in mature maize grains [13]. Another concern is the potential impact on human health due to the possible translocation of applied NPs and their residues, which could surpass cellular barriers and accumulated in edible parts of the plant [14], posing risks of accumulation and toxicity for humans. Insoluble Au-NPs [14], relatively biologically stable SiO_2_ [15,16,17], or mechanically–chemically stable TiO_2_-NPs [18] hold a potential toxicological risk in comparison to more soluble micronutrient based ZnO-NPs [19].

One of the crops that has been relatively unexplored but is attractive in the academic literature concerning the application of NPs is maize (*Zea mays* L.). It is the third-most popular crop worldwide for human consumption and the most extensively cultivated cereal [20]. Maize is an annual crop belonging to the grass family (*Poaceae*) and is recognized for its adaptability and broad food utility. Its diverse applications are demonstrated by its numerous varieties [21] with variable technological solutions [13,22].

In terms of nutrient intake, absorption, assimilation, and their impact on human health, a balanced diet that includes both animal and plant-based foods is fundamental for the basic functions of the body [23]. The mineral nutrient profile is particularly crucial in this context. However, the content and bioavailability of these nutrients in plant-based diets often lack balance. For instance, the presence of P in the diet is crucial for muscle function, cellular energy reactions, and the overall development of the organism. On the other hand, when ingested in the form of phytic acid, it binds major divalent cations such as Ca, Mg, Mn, Cu, Zn, or K, hindering their absorption and positive effects [24]. Potassium is a vital element in the human body, where it plays a crucial role in the transmembrane electrochemical gradient and various metabolic functions. The recommended daily intake varies with age and gender, ranging from 2.6 g to 3.6 g for adults. Maize grains contain more potassium than other cereals [25]. Calcium, similar to P, is crucial for the proper functioning of the nervous system, and its deficiency can lead to an increased risk of osteoporosis and weak bones. Conversely, excessive intake, when binding to insoluble oxalates, can lead to the formation of kidney stones or interfere with the absorption of other nutrients [26]. Zn is essential for the proper functioning of the immune system, cell growth, and DNA development [27] and plays a key role in plants, such as participation in over 300 enzymes in metabolism, regulation of auxin synthesis, or reinforcement of cell membrane integrity [28]. Iron is essential for hemoglobin production, and its deficiency can lead to anemia, weakness, or reduced immune function [29]. Zinc has a maximum recommended daily intake of 11 mg, while for iron, it is 18 mg [30].

Sulfur and nitrogen are essential macro-elements and fundamental building blocks of proteins [31]. The content of sulfur and nitrogen in the form of amino acids, especially methionine, is crucial for the body’s protein synthesis, proper growth, and individual development [32]. Maize is a significant source of starch, a polysaccharide that breaks down into glucose in the digestive tract [13]. Appropriate intake and concentrations of Si are crucial for bone and tissue formation and reinforcement, including the skin, hair, and nails. However, in NM form, Si may pose a human health risk [17]. Silicon content in plants can vary depending on the type of plant species or plant parts. In plants, beneficial responses have been observed following the application of Si-NPs to the soil. These responses include improvements in maize plant physiology and growth, enhanced environmental stress response [16], or antifungal properties against phytopathogens [15].

In the case of agronomic application of Ti or Au, no plant essentiality is assumed, and there is no expected positive response or function towards humans. Currently, these elements are biocompatible and are utilized in sensing applications [33] and medical implants [34]. The presence of Au in plants is most commonly mentioned in the form of the cation Au^3+^, which can negatively alter fundamental plant functions at the molecular level [3]. Additionally, Au-NPs can serve as tracers to monitor uptake, transport, and accumulation in plant structures [35]. The function of titanium in the form of cations is not fully clear in plants, but it appears that at low concentrations, it reduces stress responses, intensifies physiology, or improves yields [36].

For this reason, our goal was to apply selected types of NFs based on Au with Si, Ti, and Zn as progressive biostimulants to assess quantitative and qualitative parameters, mineral nutrient profile, and potential risk of NP translocation towards maize final quality in authentic agronomic conditions during the vegetation season of 2021 in the Central European region.

## 2. Materials and Methods

### 2.1. Origin and Characterization of Sprayed Nanoparticles Applied on Maize

Titanium dioxide (TiO_2_) and zinc oxide (ZnO) NPs were applied as foliar dispersion for maize production at a concentration of 10 mg∙L^−1^, sourced from Sigma-Aldrich (SaintLouis, MO, USA). The TiO_2_-NPs exhibited a size of approximately 30 nm, with an anatase–rutile structure and predominantly spherical morphology [8]. The physicochemical characteristics of ZnO-NPs revealed a wurtzite-type structure, predominantly spherical or hexagonal morphology, higher crystallinity [10], colloidal properties with a hydrodynamic dimension of ~280 nm, and a zeta-potential corresponding to −33.3 mV [11].

The applied Au and Si correspond to low concentrations of 0.1 mg∙L^−1^ and 10 mg∙L^−1^, respectively. Regarding the production of Au NPs, a biotechnological protocol was employed for their synthesis, involving interactions between cultured mesoporous algae cells of *Mallomonas kalinae* (SiO_2_) and ionically soluble forms of Au^3+^. This hybrid composite has already demonstrated photocatalytic efficiency [37] and an agronomically beneficial response [7].

The morphology of three NPs was investigated using the scanning transmission electron microscope (STEM) JEOL JSM-7610F Plus, Tokyo, Japan, with a Schottky cathode at 30 keV in high vacuum chamber. Samples were placed on copper grids with formvar. Elemental composition analysis and mapping were performed using the energy-dispersive X-ray spectroscope (EDS) AZtec Ultima Max 65 (Oxford Instruments, Abingdon, UK).

### 2.2. Plant Material

For the field experiment, maize (*Zea mays* L.) corresponding to FAO 360 was utilized, characterized as the “dent” type-flint. The variety Estevio exhibits rapid spring growth, fertile ears with 16–18 rows of grains, and capable of high yields in dry climatic conditions. Additionally, it tolerates stress, overheating, and withstands heavy storms. It features resilient stalks against various diseases and lodging. Under ideal pollination conditions, it forms ears with 36–38 grains in a single row [38].

### 2.3. Experimental Location Description

The site selected for real field experiments is Dolná Malanta, near Nitra (N 48°19′25.41″ E 18°09′2.89″), Slovak Republic, Central Europe. From an agronomic perspective, it is a crop region characterized by maize production, where intensive soil cultivation occurs [39]. Geographically, it is situated at an elevation of 250 m above sea level and is located in the Danubian Lowland, bordered to the north by the Tribeč mountain range and to the east by the Žitavská Hillock. Geologically, the site is situated in the Western Carpathians, characterized by composite leucocratic granites and carbonate rocks with Mesozoic development. Various types of eluvial sediments with Neogene and Quaternary development are stratigraphically overlain on the bedrock [40], while the experimental soils correspond to silt loam haplic Luvisols [41]. Soil characteristics are depicted in Table 1. The soil reaction is weakly acidic, with an average acid exchange soil reaction and very strong hydrolytic acidity. The soil sorption complex exhibits a medium to high total sorption capacity, as evaluated using the Kappen’s method, and is saturated to fully saturated with basic cations [42]. Other soil characteristics are shown in Table 1.

The TOC exhibited a moderate value according to methodology settled by Ťurin, with labile carbon values averaging 1761.4 ± 102 mg·kg^−1^. Determination of humic substances were used by Kononova and Bělčikova [43], where HS correspond to averaged 0.49%, with the extractability of HA at 0.232 ± 0.02% and FA at 0.93%. The ratio of HA/FA ranged from 0.78 to 1.02 (%), with an average close to 1. Content of the carbonate content (CaCO_3_^2−^) was low with unified protocol diluted by water ratio HCl 1:3 [42]., and the EC values varied around 174.67 ± 9.90 μS·cm^−1^.

### 2.4. Seasonal Fluctulation of Air Temperature and Precipitation during Vegetation Season 2021

To analyze seasonal variations, the meteorological station at the Slovak University of Agriculture in Nitra conducted a monitoring program to track precipitation and average monthly temperatures during the 2021 growing season. Results were compared with long-term climatic norms (Figure 1a,b).

### 2.5. Field Experiment

The field experiment was conducted in a kind of multifactorial experiment using orthogonal divided plots [44], replicated three times with randomly arranged experimental units. The total experimental area measured 1242 m^2^ (27 m × 46 m), with each individual experimental plot (replication) sized at 60 m^2^ (6 m × 10 m). Maize cultivation was carried out conventionally with standard agronomic practices, including deep plowing using a Zetor tractor 6211 (Zetor Tractors, a.s., Brno, Czech Republic) in autumn 2020 and pre-sowing preparations in spring 2021. Maize was grown in a 7-plot crop rotation system, with sugar beet (*Beta vulgaris* sk. Altissima) preceding it as the pre-crop.

The NPK fertilizers were applied to the soil via tractor-mounted application equipment during pre-sowing soil cultivation utilizing a Ferti fertilizer applicator (Agromehanica, Boljevac, Serbia). The NPK 15-15-15 DUSLOFERT fertilizer (Duslo, a. s., Šaľa, Slovakia) was dosed at 450 kg·ha^−1^. Application rates were determined based on agrochemical analyses conducted prior to maize sowing, reflecting standard concentrations according to Slovak legislative norms [45].

Maize seeds were sown in rows at a depth of 70 mm with a seed spacing of 220 mm and inter-row spacing of 700 mm using a Monosem NG Plus 3 planter (Monosem, Largeasse, France). Following planting, a pre-emergent application of the herbicide WingP^®^ (BASF, Ludwigshafen am Rhein, Germany) at a concentration of 4 L∙ha^−1^ was applied two days after sowing. All treatment variants, including the control, were evenly dispersed onto maize leaves using an AGT 865T/S sprayer (Agromehanica, Boljevac, Serbia).

Water-dispersible solutions were applied onto the leaves of the plants using a hand-held sprayer (Mythos Di Martino, Mussolente, Italy). Application was conducted early in the morning under calm weather conditions until the maize leaves were thoroughly wetted. The first application of NPs solution occurred 25 days after sowing at the five-leaf stage–ear and tassel initiation, while the second application was carried out 50 days after planting during the twelve leaf “ear size determined” growth phase.

In the control treatment, only water was utilized for preparation. Likewise, the same water was employed for the preparation of colloidal NP solutions, which were subsequently deposited on the maize leaves, similar to Ernst et al. [7]. Prior to each application, the NPS were perfectly dispersed using ultrasonic treatment.

### 2.6. Evaluation of Yield, and Yield Components of Maize

All relevant quantitative parameters of maize were analyzed in the Laboratory of Quantitative Analyses at the Institute of Agronomic Sciences, Faculty of Agrobiology and Food Resources, Slovak University of Agriculture in Nitra. The number of plants and the number of ears were determined per unit area in units of plants per hectare during pre-harvest inventory.

Ear length, ear diameter, and cob diameter were evaluated using standard methods employing a Texi 4007 device (Texi GmbH, Berlin, Germany); the number of rows and grains per row were also assessed. Grain weight from the ear, 1000-grain weight, and cob weight were analyzed using a laboratory scale of Kern type KPZ-2-05-3 (KERN & Sohn GmbH, Balingen, Germany), supplemented by seed counting using a laboratory seed counter Numirex (MEZOS spol. s.r.o., Hradec Králové, Czech Republic).

### 2.7. Evaluation and Nutritional Parameters Profile of Maize with Potential Translocation of NPs or Their Residues

Maize kernels, at full maturity, underwent analysis for selected mineral nutrients and starch content. Nitrogen was analyzed using the Kjeldahl method corresponding to crude protein, and sulfur content was determined colorimetrically. Phosphorus content was determined spectrophotometrically, while potassium was analyzed by flame photometry [46]. Zinc, calcium, and iron were analyzed using F-AAS (Perkin-Elmer Model 1100, Waltham, MA, USA) according to the method described by Losak et al. [47]. For this purpose, seed samples weighing 0.15–0.30 g were digested in a mixture of HNO_3_ and HClO_4_ (2:1, *v*/*v*). Titanium, silicon, and gold were analyzed using ICP-MS, and sample decomposition was performed using the Anton Paar Multiwave 3000 microwave digestion system (Graz, Austria) in PTFE pressure vessels with a concentrated mixture of HNO_3_ and H_2_O_2_ at a pressure of 60 bar. Starch content was determined according to methodology Obadi et al. [13]. 

### 2.8. Evaluation of Phytoavailable Distribution of Nutrients from Soil 

The extraction of the phytoavailable fraction of macro- and micro-nutrients from soils utilized the chelating agent Melich III. This was achieved by combining 100 mL of the reagent with 10 g of soil in a 250 mL polyethylene bottle. After 10 min of shaking, the solution was filtered and subsequently analyzed colorimetrically for the content of P, K, Ca, and Mg, according to Cade-Menun et al. [48]. Similarly, soil extraction procedures employing various reagents and a shaking regime were also employed; for instance, S-content was obtained from ammonium acetate and analyzed according to the protocol outlined by Hrivňáková et al. [42]. DTPA was employed for the extraction of the micronutrient zinc, and after extraction and filtration, it was analyzed using atomic absorption spectrophotometry (AAS).

### 2.9. Statistical Operation

The obtained data were subjected to statistical analysis using the ANOVA and Fisher LSD test with a significance level set at α = 0.05 and α = 0.01. Statistical computations were performed applying with TIBCO Statistica^®^, Version 14.0 [49] (TIBCO Software Inc., Palo Alto, CA, USA).

## 3. Results

### 3.1. Physicochemical Properties of Nanoparticles Applied to Maize

The SEM imaging has shown that ZnO-NPs have a spherical shape with a rare, rod-like and cuboidal morphology with EDS verification of zinc corresponding to the stoichiometry of ZnO nanoparticles (Figure 2).

Spherical and prismatic morphologies, rarely bi-pyramids or tabular crystals are typical for TiO_2_-NPs. The EDS spectra recorded titanium corresponding to the chemical formula TiO_2_ (Figure 2c,d). Au-NPs of a predominantly spherical shape, rarely triangular or rod-shaped, with a size from 4 to 30 nm were surface-anchored on *Mallomonas kalinae* biosilica (Figure 2e,f).

### 3.2. Phytoavailability of Selected Soil Nutrients

The results (Table 2) revealed relatively low levels of phytoavailable soil content of Mg, K, P, Fe, and S, with average values for Zn and Ca.

### 3.3. Effect of Foliar Application of Metal-Based Nanoparticles on Yield Components and Grain Yield of Maize

In the study of NPs application effect on the number of plants and number of ears, all variants of NP application had observed values similar with no statistically significant change compared to the control (Table 3). Regarding the agronomic effect of the of NPs, the Au-NP-bioSi and ZnO-NPs variants exhibited the most beneficial outcomes in terms of yield at a statistically significant level and several yield-related parameters, including cob weight, cob diameter, ear length, WTS compared to TiO_2_-NPs, and the control. Other monitored parameters showed either non-significant or indicative variations across the variants. Additional parameters such as the kernel row number and number of kernels per row showed slight variations across the different treatments but without statistical significance (Table 3).

### 3.4. Evaluation of Quality and Nutritional Parameters of Maize Kernels Based on Starch Content, Mineral Nutrient Profile, and Potential Hazardous Translocation of Nanoparticles, or Their Residues

When considering the uptake and absorption of mineral nutrients, such as N, P, K, and S, that influence the final quality of maize kernels, the most efficient treatments were Au-NP-bioSi and ZnO-NPs compared to the control. The treatment with ZnO-NPs led to higher values for calcium content. The Au-NP-bioSi treatment caused the lowest calcium content and low iron content, while the ZnO-NPs induced lower concentrations of Fe. The TiO_2_-NPs treatment was less effective, but the treatment still allowed for sufficient intake of N and K (or P and Fe) at a significant level, and intake of calcium and sulfur was lower than in the control. In the case of the control, no NP treatment led to a higher intake of iron and calcium, but all other nutrient values were the lower including P, compared to other treatments.

Regarding starch content, Au-NP-bioSi and TiO_2_-NPs treatments were statistically similar to NPs-free control, while the ZnO-NPs treatment exhibited lower starch content (Table 4).

Regarding the potential hazardous translocation of catalytically active NPs, the ZnO-NP, Au-NP-bioSi, and TiO_2_-NP treatments did not increase the Zn, Au, Si, or Ti content in the final kernel and no hazardous translocation is assumed (Table 4). The control experiment had statistically the highest zinc content. In the case of gold application and potential translocation negatively influencing the final quality of maize kernels, no increased content was observed in any of the variants, with it not exceeding the detection limit of <0.002 mg∙L^−1^. Silicon reached its highest, albeit statistically insignificant, value in the control and TiO_2_-NPs variant, while paradoxically, the lowest value was observed with the application of Si-based in biocomposite form with Au-NPs. In the case of titanium content in the final kernel quality, no significant differences were observed between the variants, with the highest content observed in the ZnO-NPs variant not exceeding 0.04 mg∙kg^−1^.

## 4. Discussion

### 4.1. Assessment of Phytoavailability of Selected Nutrients from the Soil Environment for Maize Production Purposes

In the rhizosphere, mineral nutrients occur as free ions, dissolved associated ion complexes, ions absorbed onto organic soil components, inorganic precipitates, or incorporated into mineral structures [50]. Soil chemical composition, particularly the phytoavailable fraction, influences mineral nutrients uptake, such as in maize [51], thereby affecting nutritional value and final grain quality with human impact [52].

In our case, despite the low concentration of carbonates, CO_3_^2−^, (~0.2%) in soil samples (Table 1), the content and phytoavailability of Ca^2+^ may be associated with the presence of clay minerals, iron oxides and hydroxides, or organic matter–humus promoting soil aggregate formation [53], or in ionically soluble forms with nitrates or sulfates [54]. In this study, Ca had a phytoavailable concentration of around 1600 mg∙kg^−1^, which should be considered sufficient for plant root systems in the rhizosphere [55]. Ca deficiency is predominantly associated with a low pH, leading to limited meristem cell development. Ca^2+^ and Mg^2+^ exhibit competitive relationships for binding sites in the soil environment [56], and during Mg deficiency, the substitution of Ca for K in maize leaves may appear [57]. Magnesium deficiency can reduce photosynthetic efficiency [56]. The phytoavailable concentration of Mg at 278 mg∙kg^−1^ was lower compared to Ca and general soil values considered suitable for growing (Table 2) [54]. Usually, Mg^2+^ is an inorganic mineral that forms in soil and it is present in natural organic matter. In soil solution, Mg^2+^ is primarily taken up and absorbed by the maize root system through ion exchange mechanisms [56].

In our case, the K bioaccessible concentration corresponds to ~212 mg∙kg^−1^, which is relatively low, as its values for uptake by plants can reach up to 10,000 mg∙kg^−1^ [58]. Furthermore, K can be sorbed on clay surfaces and thus easily accessible; it also forms water-soluble complexes or organic substances [59]. The uptake of K by roots occurs through specific proteins and is redistributed by the plant’s active transport system [55]. The soil contained a low phytoavailable concentration of phosphorus-25 mg∙kg^−1^. Most of the time, P occurs in both organic and inorganic forms [60], with most biologically available forms, such as H_2_PO_4_^−^ and HPO_4_^2−^ [55]. The phytoavailable sulfur in the soil primarily exists as (SO_4_^2−^), easily soluble and accessible in soil solution environments [55], which naturally crystallizes into solid forms with elements like Mg, Ca, and K [61].

The concentration of phytoavailable zinc in soils was relatively low (~2.6 mg∙kg^−1^), with no apparent occurrence in the soil–inorganic part, although our concentrations range was higher than in the zinc-deficient soils in North America [62]. A higher ratio of exchangeable Mg:Ca was not demonstrated in our soils, indicating an even greater potential deficit of Zn [27]. Moreover, in the case of maize, Zn availability may be limited by antagonistic relationships with phosphorus [63]. Generally, zinc is commonly integrated within mineral structures, where it substitutes for other divalent cations, such as calcium [27]. For maize, zinc is typically supplied through foliar application, a strategy currently used for its targeted effect in zinc-deficient soils [64,65]. In our case, the ZnO-NPs benefits were already determined in the case of sunflower [66], lentil [11], and foxtail millet [10]. Zinc can also occur in organic form or absorbed, for example, onto clay minerals, primarily in geochemical distribution of Zn^2+^. In its deficiency, several undesirable growth changes may appear, such as chlorosis, plant deformities, or developmental disorders [28].

The iron-phytoavailable forms in soil had a concentration at 13.5 mg∙kg^−1^ (Table 4). Iron can occur in oxides and hydroxides, weathered mafic minerals, such as biotite and amphibole, chelate forms, such as siderophores, or in organic matter. Iron deficiency or its immobilization is usually associated with soils containing high amount of carbonates that have a high pH [67].

### 4.2. Effects of Micronutrient-Based and Non-Essential Nano-Fertilizers on Yield Components and Grain Yield for Maize

In terms of the number of plants and ears, we found no statistically significant difference across all treatments, including the control. The applied concentration of 10 mg∙L^−1^ was relatively ideal for maintaining maize morphology without leaf deformation or chlorosis initiation, potentially avoiding reduced yield. In this context, Ehsanullah et al. [68] observed a higher number of plants per hectare with foliar application of conventional ZnSO_4_ across multiple maize hybrids. This is probably due to the synergistic effect with nitrogen, which correlates with foliar zinc application because its translocation occurs during the grain development from older leaves to younger ones. Our study aims to first confirm it in the Central European region under real agronomic conditions, with a positive effect to parameters including WTS and cob diameter (Table 3). Umar et al. [65] observed the intensification of all monitored aboveground and root growth parameters with enhanced chlorophyll value and photosynthetic rate of maize upon the ZnO-NPs application and conventional soluble-ionic form of ZnSO_4_, discussing the most available species of Zn for foliar application from ZnO-NPs. Moreover, both soil and foliar applications of ZnO-NPs contributed to the highest yield, somewhat similar to our results (Table 3).

In the context of yield components and gain yield, the most effective treatment was observed with Au-NP-bioSi. This is potentially due to the efficient delivery of phytoavailable Si, which likely enhances resistance against pests such as fungal pathogens, including *Fusarium oxysporum* and *Aspergillus niger* [15]. Consequently, the crop exhibits improved health and potentially enhanced yield production, which was observed in our experiment as well. Soil-applied nano-silica induced physiological changes in maize, positively stimulating the expression of organic compounds such as chlorophyll, proteins, and phenols, including quantitative parameters such as dry weight and shoot growth [16], which could also explain the enhancements observed in our yield components. In the soil environment, Au-NP-bioSi could accelerate the availability of soil nutrients and other processes such as N-fixation, P-solubility, and microbial diversity compared to micro-silica in the rhizosphere during maize cultivation [69]. Furthermore, maize requires relatively high amounts of light energy and heat [21]. A hybrid mesoporous silica (SiO_2_) exhibited improved light absorption and chlorophyll *a* and *b* photosynthetic activity [70]. Most likely, this results in more efficient photosynthesis in maize, associated with higher yields and yield components (ear length, kernel row number, and WTG) (Table 3). The association of intensified photosynthesis with higher yields in the case of Au-NP-bioSi and ZnO-NPs has already been demonstrated for sunflower at the same experimental location [7]. An effective yield response and quantitative parameters may also be caused by the synergistic effect of Au-NPs and Si, a concept so far absent in the agronomic literature regarding foliar deposition on maize.

The use of TiO_2_-NPs as a plant stimulator did not yield significant benefits in maize compared to sunflower [7]. Although some yield components such as ear length, kernel row number, and TWS showed improvement over the control, the differences were not statistically significant (Table 3). Titanium, as an element, is relatively unavailable and inadequately mobilizable, and its diffusion within the plant is limited, with most of its ionic-organic species being somewhat metabolizable [71]. Thus, the foliar application of inorganic rutile–anatase TiO_2_-NPs modification in sizes of about 30 nm only had a weak impact on yield intensification in this study. In a study by Hussain et al. [72], spray application of the Ti in ionic form on soybean showed a beneficial effect on photosynthesis, chlorophyll content, leaf area, P uptake, and grain yield, which partially aligns with our findings.

### 4.3. Assessment of Maize Grain Quality Based on the Content of Mineral Nutrients Profile and Starch

In general, maize kernels contain approximately 65–72% starch, ~10% proteins, 3.5–5% oil, 1.4–2% sugars, and 1–2.5% ash [29,32]. Grain quality factors mainly address the chemical and physical properties of mature grains, specifically kernel size, uniformity and weight, wet milling [32], starch extraction [73], fiber content, sugars, oils [74,75], nutritional minerals, such as iron and zinc [29], vitamins A, E, and antioxidants [31]. Our cultivar is appropriate for immediate consumption, with additional assessments possible to determine the glycemic index and the presence of antinutritional compounds, including P content [24].

All treatments with NFs demonstrated higher P content in the final grain quality compared to the control (Table 4). However, the P content reached standard threshold values in maize grains [31,75]. However, when assessing its potential food quality, it indirectly indicates the presence of phytic acid [24], which is less desirable.

All NP treatments led to higher concentrations of K in the grain of maize (Table 3). Even though higher K concentrations in the entire plant and subsequently in the grain lead to higher production of sugars influencing cooking quality, this was not visible in the relative increase of starch in grains of maize and only in higher yields in the case of ZnO-NP and Au-NP-bioSi treatments. TiO_2_-NP treatment and subsequent higher K concentration in grains, led to slightly higher levels of starch in the grain, but not to significantly higher yields. In terms of the highest starch content, the most effective were TiO_2_-NP and Au-NP-bioSi treatments followed by control treatment while the ZnO-NPs treatment resulted in the lowest values (Table 3). Starch formation in plants depends on complex processes involving multiple factors. Starch synthesis is not only dependent on phytoavailable K, but also other macro-nutrients, such as N, P, Ca, and Mg, as well as micronutrients, such as Fe, Mn, Zn, Cu, which also play a crucial role in transient starch quality, storage starch synthesis, and cooking quality [76]. Another critical factor influencing sugar followed by starch formation is the variability of seasonal fluctuation [75]. During its key vegetative and reproductive phase of development, the synthesis and accumulation of sugars were relatively optimal in terms of temperatures based on monthly climatic normal (Figure 1a), but less favorable in terms of precipitation compared to the long-term climatic normal (Figure 1b). It is surprising that when comparing treatments, agronomically unconventional biostimulators, such as Au-NP-bioSi and TiO_2_-NPs with a control, had increased starch content compared to the ZnO-NPs micro-based NFs. Additionally, Zn, which should have activated enzymes involved in sugar formation and photosynthesis in chloroplasts, transports from leaves to seeds, which, through enzymatic reactions, transform into starch and fill grains, defining their final quality [27]. Our results show that higher yields of maize grains in ZnO-NP treatment led to a slightly lower content of starch. The concentrations of Zn we applied, thus, were only for 5.7% higher yields with 3.6% lower starch content (Table 4). The slightly highest yield, yield components, cob diameter, WTS, and cob weight, and starch content were found in the non-essential Au-NP-bioSi treatment (Table 3 and Table 4). This may be related to more intense photosynthetic reactions caused by the absorption of Si, which helps with the attenuation of ROS activity in the plant, or, maybe nanospecific effects of Au-NPs, as was observed in sunflower [7]. Here, sunflower seed quality is based on the essential acids where the higher proportion of essential linoleic acid and oils extraction correspond to the Au-NP-bioSi variant.

In economically disadvantaged regions of the world, maize serves as one of the main sources of Fe. According to [77], the daily intake of Fe for humans should range from 8 to 11 mg. In our case, the highest concentrations were recorded for the NPs-free control and all other treatments contained lower concentrations of Fe that were comparable among the treatments with NPs. This lower concentration of Fe by 13 to 18% is not a desirable effect with NP treatments. However, this can be viewed as the dilution effect of higher yields (Table 3).

Additionally, besides Fe, the control is also the most suitable for the highest concentrations of Zn. Several agronomic strategies aim to increase its levels through biofortification, one of them is also a foliar application of ZnO-NPs [65]. Our applied concentrations of ZnO-NPs were sufficient to increase the yields of maize, but, unfortunately, not the Zn concentration in grains.

Generally, N content is used as an indicator of crude protein in maize grains [31] or as a measure of nutritional value for animal feed, where it is a key element for amino acid synthesis, thus serving as the proteins building block [32]. The N in mature maize grains had higher concentrations across all NFs treatments (Table 4). The highest values were observed for Au-NP-bioSi and ZnO-NPs, with slightly lower levels for TiO_2_-NPs compared to the lower values in the NP-free control. A similar situation occurred in the evaluation of sulfur, where approximately double concentration was found in the Au-NP-bioSi and ZnO-NPs variants compared to the control and TiO_2_-NP treatments. Content of N is primarily associated with maize proteins, although mostly in biologically less valuable amino acids, namely lysine and tryptophan [32]. Apart from zein, maize also contains globulins and albumins with a wider spectrum of amino acids, including organic sulfur, albeit in smaller quantities. Our results may thus indicate that the maize grains contained a higher quantity of the quality-enhancing glutelins containing cysteine and methionine [75] after the application of ZnO-NPs and Au-NP-bioSi, where higher levels of sulfur were recorded.

The generally known negative relationship between yield and nitrogen integrated into grain proteins [32] was not observed in our case, which could be attributed to the application of unconventional NFs. It is worth noting that the maize cultivation had a relatively sufficient supply of available nutrients. The availability of N, P, and K in the soil is a key factor for proper growth, adaptation, and production of maize protein, as well as for starch and oil production [32,74,76]. In our conditions, P and K were available in relatively sufficient concentrations, although S was somewhat deficient (Table 2) [61]. Suitable soil conditions were also important, including a pH range of ~6.1 at our location, high saturation of the sorption complex with basic cations, and low carbonate content, although the content of organic matter and humus was rather low (Table 1). As for the starch content and S content in maize grains, the highest values were observed in the Au-NP-bioSi treatment, along with the ZnO-NPs treatment, while the content was lower in the TiO_2_-NPs treatment and in the control.

### 4.4. Assessment of the Potential Risk of Translocation of Applied Metallic Nanoparticles or Their Residues Affecting the Final Quality of Maize Grains

In the case of ZnO-NPs, no increase in its content was observed in maize grains the influencing quality compared to other variants involving control (Table 4). Logically, this implies that ZnO-NP uptake by the plant or uptake of ionic Zn release from the NPs were linked to the metabolic and physiological maize processes, as stated by Srivastav et al. [64] in the case of multiple soil applications with higher doses than our foliar application. Most likely, the supplied Zn role had a non-toxic, beneficial role that resulted in higher yields and yield components (Table 3), and the lower content in grains indicates a dilution effect. The relatively lower content in maize grains could have been caused by low applied concentrations of 10 mg∙L^−1^. Potential translocation and intensified transfer and accumulation into maize grains at higher concentrations of ZnO-NPs have not yet been documented in the academic literature. Even though in our case the application of ZnO-NPs leads to a 22% lower concentration, the content of Zn in grains aligns with standard values associated with dry mass of maize grains [78]. After foliar application, ZnO-NPs on maize may transform and distribute into a zinc-soluble form, similarly like in sunflower leaves [79], potentially restricting their transfer into maize grains. The optimal daily dose of Zn for humans’ ranges from 8 to 11 mg for adults [27]. In the hybrid used in our study, the concentration of Zn in them can only contribute to a partial dose of the Zn daily intake compared to some Zn-rich hybrids with the potential for higher intake and accumulation when consuming equivalent quantities [27].

Regarding the application of gold-based NPs, no higher content was observed in the maize grains (at the detection limit of analysis of <0.002 mg∙L^−1^). It is considered relatively biocompatible, but depending on size, charge, and surface modification upon entry into the human body, it may cause potentially genotoxic changes at the cellular level [10,14]. Examination of cellular uptake, permeability, biodistribution, and systematic adsorption of Au-NPs revealed relatively low levels of toxicity in orally ingested rats; however, it can still be considered relatively cytotoxic [80]. Gold is non-essential for humans, but from an agronomic perspective, it is classified as a plant biostimulant at low concentrations [3], which, when loaded on biosilica, proved to be most effective in promoting the yield in our study (Table 3). Moreover, in terms of physiological stimulation effects in our locality, it has already demonstrated its beneficial role in sunflower cultivation [7], where no hazardous translocation affecting the final quality of sunflower seeds was observed.

The Au-NPs were anchored to the *Mallomonas* biosilica hybrid composite, which is considered to be a more biocompatible and bioavailable form of Si than its pure crystalline forms [17]. In this case as well, the mechanism of action remains unclear, with silica not being observed in higher concentrations in the maize grains (Table 4). Conversely, Si translocation was evident in the sunflower treatment, where the Au-NP-bioSi treatment led to double the concentration compared to the NP-free control [7]. Most probably, the phytoavailable Si was metabolized and used in the formation of plant tissues, such as leaves during the formation of phytoliths, trichomes, ear, or supporting the formation of phytoplasmic bodies or mechanical structures in plants, where Si is preferentially accumulated in higher concentrations [70]. The recommended Si daily intake for humans should not exceed 700 mg for adults [81], and the Si intake in NP forms may be potentially hazardous [17].

In terms of the transfer of TiO_2_-NPs or its potential residues, approximately similar titanium contents were observed in the final grains in all treatments, ranging from 0.03 to 0.04 mg∙kg^−1^ (Table 4), which can be considered low concentrations. TiO_2_-NPs were in the form of crystalline anatase–rutile NPs, which are chemically inert and mechanically stable. However, TiO-NP transfer may have several adverse effects on the environment, including its potential carcinogenicity to humans [18]. Due to its physicochemical properties, non-essential nature, and toxicity, titanium, particularly in nano-forms, was classified as potentially hazardous by the European Food Safety Agency in 2008 [18]. Moreover, Ti is agronomically classified as a growth stimulator with unclear physiological effects on plants. At low concentrations, especially when sprayed as a dispersion, it provokes several positive agronomic changes, for example, in sunflowers [7,8], or intensifies the quantity and quality of essential oils and the biosynthesis of secondary metabolites in feverfew [82].

## 5. Conclusions

The most beneficial for yield and yield components of maize in foliar application under authentic agronomic conditions of the Central European region were the variants of Au-NP-bioSi and ZnO-NPs on the statistical level, as opposed to TiO_2_-NPs and NPs-free control for most of the observed parameters. The relatively appropriate phytoavailability of soil macro- and micronutrients supported their uptake and their content in the final grain quality of maize. Here, the lowest concentration of the antinutrient phosphorus was found in the NPs-free control; the indication of optimal crude protein content, based on higher N and S content, was associated with Au-NP-bioSi and ZnO-NPs; and the starch content ranked as follows: TiO_2_-NPs > Au-NP-bioSi > control > ZnO-NPs. It can be summarized that the final quality of the maize grains was not affected by the translocation of any of the applied NPs, especially the stable Au-NP-bioSi and TiO_2_-NPs. Our results favor the application of low concentrations of NPs as an effective agronomic tool for precise and sustainable agriculture without unintended side effects.

## Figures and Tables

**Figure 1 plants-13-01936-f001:**
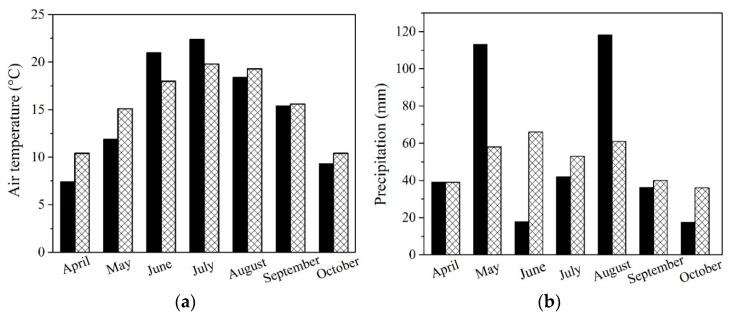
Analyzation of the monthly variations in (**a**) air temperature and (**b**) precipitation at the experimental field site in Dolná Malanta near Nitra, Slovakia, within the Central European region. Dark black columns represent measured data for the vegetation season compared to the ideal climatic norm (sparse texture) for both cases.

**Figure 2 plants-13-01936-f002:**
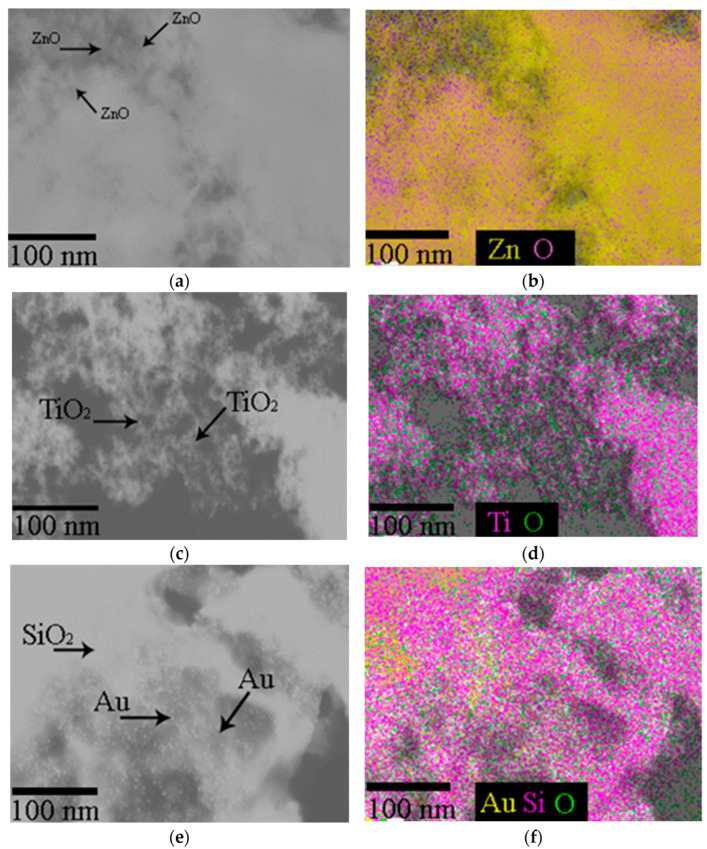
(**a**,**b**) Images of zinc oxide (ZnO-NPs); (**c**,**d**) titanium dioxide (TiO_2_-NPs) NPs; (**e**,**f**) Au NPs anchored to meso-biosilica (Au-NP-bioSi) that were applied as a spray dispersion, visualized by scanning electron microscope and chemical map verification with EDS.

**Table 1 plants-13-01936-t001:** Initial soil parameter before sowing in the vegetative season of 2021 at the experimental site: Dolná Malanta, Slovakia, Central Europe.

Soil Parameters	Average Value with Standard Deviation
pH _H_2_O_	6.118 ± 0.44
pH _KCl_	5.341 ± 0.36
Hydrolytic acidity (mmol·kg^−1^)	18.296 ± 5.18
Exchanable basic cations (mmol·kg^−1^)	172.07 ± 23.80
Total sorption capacity (mmol·kg^−1^)	190.37 ± 18.63
Sorption saturation level of basic cations (%)	90.21 ± 3.68
Total organic carbon (TOC) (%)	1.381 ± 0.10
Humic substances (HS) (%)	0.493 ± 0.005
Humic acid (HA) (%)	0.232 ± 0.02
Fulvic acids (FA) (%)	0.261 ± 0.02
Ratio of HA/FA	0.93 ± 0.17
Carbonate content, CaCO_3_^2−^ (%)	0.19 ± 0.01
Electrical conductivity, (EC) (μS·cm^−1^)	174.67 ± 9.90
Soil texture	
Clay (%)	15
Silt (%)	49
Sand (%)	36

**Table 2 plants-13-01936-t002:** Content of phytoavailable macro- and micro-nutrients (in mg·kg^−1^) from soil prior to sowing in the vegetative season of 2021 at the experimental locality Dolná Malanta, Slovakia, Central Europe.

Mg	Ca	K	P	S	Zn	Fe
278.45 ± 6.93	1600 ± 70.71	212.5 ± 3.54	25 ± 0.04	3.75 ± 0.002	2.67 ± 0.05	13.51 ± 0.03

**Table 3 plants-13-01936-t003:** Comparison of yield components and grain yield of NPs-free control ZnO-NPs, Au-NP-bioSi, and TiO_2_-NPs treatments recorded in the 2021 vegetation season.

	Control	ZnO-NPs	Au-NP-bioSi	TiO_2_-NPs
	Quantitative Parameters			
Number of plants per hectare (thousand per ha^−1^)	73.24 ± 0.11	73.27 ± 0.07 ^ns^	73.16 ± 0.19 ^ns^	73.29 ± 0.13 ^ns^
Number of ear (thousand per ha^−1^)	109.87 ± 0.16	109.95 ± 0.67 ^ns^	109.76 ± 0.46 ^ns^	109.99 ± 0.36 ^ns^
Ear length (mm)	185 ± 18	208 ± 6 ^ns^	213 ± 9 *	199 ± 17 ^ns^
Ear diameter (mm)	47 ± 3	50 ± 5 ^ns^	51 ± 3 ^ns^	47 ± 5 ^ns^
Kernel row number (pcs)	19 ± 1	19 ± 3 ^ns^	21 ± 1 ^ns^	19 ± 3 ^ns^
Kernel number per row (pcs)	39 ± 3	39 ± 3 ^ns^	36 ± 4 ^ns^	36 ± 4 ^ns^
Weight of grain per ear (g)	185.67 ± 12.36	218.60 ± 49.32 ^ns^	238.50 ± 17.37 ^ns^	201.17 ± 47.92 ^ns^
Weight of thousand seeds (WTS) (g)	248.82 ± 14.25	291.43 ± 20.22 **	308.70 ± 11.80 **	262.93 ± 17.64 ^ns^
Cob weight (g)	20.93 ± 2.26	26.57 ± 5.77 ^ns^	29.83 ± 1.16 *	21.80 ± 6.20 ^ns^
Cob diameter (mm)	23 ± 1	27 ± 3 *	26 ± 1 ^ns^	22 ± 2 ^ns^
Total number of kernels per ear (pcs)	748 ± 82	745 ± 119 ^ns^	751 ± 97 ^ns^	705 ± 157 ^ns^
Grain yield (t·ha^−1^)	7.86 ± 0.31	8.31 ± 0.19 *	9.15 ± 0.27 **	8.13 ± 0.14 ^ns^

The significance: * *p* value < 0.05, ** *p* value < 0.01, ^ns^ non-significant. The values present after symbol ± show the standard deviation.

**Table 4 plants-13-01936-t004:** Profile of selected mineral nutrients, nutritional parameter–starch, as well as potentially translocated elements or their residues into mature maize grains (mg∙kg^−1^) during the 2021 growing season at the experimental site: Dolná Malanta, Slovakia, Central Europe.

	Control	ZnO-NPs	Au-NP-bioSi	TiO_2_-NPs
Mineral Nutrients (mg kg^−1^)				
Nitrogen	9054 ± 108.2	12,084 ± 219.2 **	12,439 ± 439.8 **	10,679 ± 219.2 **
Phosphorus	2192 ± 14.5	2697 ± 97.6 ^ns^	3193 ± 196.6 **	2375 ± 353.6 ^ns^
Potassium	3288 ± 29.4	3651 ± 156.3 ^ns^	3859.5 ± 195.9 *	3648 ± 77.8 ^ns^
Calcium	259.5 ± 20.5	249 ± 39.59 ^ns^	180.5 ± 19.1 ^ns^	193.5 ± 38.89 ^ns^
Sulphur	925 ± 145.7	1521.5 ± 195.87 **	1701 ± 49.5 **	829 ± 12.9 ^ns^
Iron	49.7 ± 2.26	40.7 ± 2.4 *	42.6 ± 2.26 ^ns^	43.4 ± 3.39 ^ns^
Nutritional Parameter (%)				
Starch content	65.29 ± 0.13	62.94 ± 0.26 **	65.57 ± 0.05 ^ns^	66.42 ± 0.13 **
Potential Translocated Elements (mg kg^−1^)				
Zinc	30.4 ± 0.42	23.7 ± 1.98 **	25.4 ± 2.54 **	23.25 ± 1.63 **
Gold	>0.5	>0.5 ^ns^	>0.5 ^ns^	>0.5 ^ns^
Silica	24.92 ± 1.25	20.89 ± 1.05 *	16.98 ± 0.85 **	22.86 ± 1.14 ^ns^
Titanium	>0.03	0.04 ± 0.001 ^ns^	>0.03 ^ns^	>0.03 ^ns^

The significance: * *p* value < 0.05, ** *p* value < 0.01, ^ns^ Non-significant. The values present after symbol ± show the standard deviation.

## Data Availability

Data is contained within the article.

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
