# Peer review of "Enhancing Maize Yield and Quality with Metal-Based Nanoparticles without Translocation Risks: A Brief Field Study"

_plants, 2024, doi:10.3390/plants13141936_

Round 1

Reviewer 1 Report

Comments and Suggestions for Authors

Comments

Before delving into specifics, I'd like to begin with a general comment. This manuscript explores the impact of foliar treatments using Au-NPs on maize. Early studies on the interaction between nanoparticles (NPs) and plants involved experiments with Ag-NPs, Au-NPs, and other metal-based NPs in hydroponic culture to assess root uptake and their distribution within plants.

Fast forward 20 years, and we find ourselves in the era of "nano-enabled agriculture". Therefore, it's essential for us to immerse ourselves in a milieu of research and experimentation under real field conditions. While I understand that the authors have already gained experience with the benefits of treatments involving Au-NPs under controlled conditions, I am curious if the authors believe they can execute treatments based on Au-NPs on a large scale. How much would this material cost? What would be the sources of supply? Additionally, what price point for corn grain would ensure a profit for farmers after covering production costs?

Low Nutrient Use Efficiency, i.e., low fertilisation efficiency globally, has high economic and environmental costs. From this point of view, introducing nanofertilizers could bring significant benefits. However, it is crucial to consider the potential scenario where the high costs associated with these innovations could lead to a situation where nanofertilizers are only accessible in richer countries. This could potentially exclude poorer nations from reaping their benefits, highlighting the urgent need for strategies to ensure equitable access.

While it's important not to stifle the progress of research, we must always remember that scientific research is not an end in itself. It's crucial to maintain a keen focus on reality, understanding the real-world implications of our findings, and the potential impact they can have on global agriculture and environmental sustainability.

1. Introduction

1.1) rows 42-44: “Some pioneering studies focus on applications to enhance crop production using “exotic” materials, such as precious metals like Au, or Si- or Ti-based materials [1]”.

This statement is wrong. In the review cited on row 44 (Liu and Lal. 2015. STOTEN 514, 131-139. https://doi.org/10.1016/j.scitotenv.2015.01.104), the use of nAu is not mentioned. I suggest reformulating the sentence or, even better, supporting it with correct references.

1.2) rows 82-125: This section of the introduction is useless. Arguing on the physiological role of macronutrients in plant nutrition and the consequences on humans of a diet deficient/or excessively rich in the same elements is only helpful for filling lines of text.

For example: rows 100-101: Excessive doses (of Zn) are associated with nausea, vomiting or weakened immune system function

Delete this part, including the dozen or so literature quotes.

2. Materials and Methods

 a) paragraph 2.3 - Experimental location description

Delete rows 172-178, not relevant. Insert data regarding soil texture (% sand, lime, clay) in Table 1.

b) paragraph 2.4 – Climatic seasonal variations during vegetation season 2021.

Apart from the SEM images, the only figure presented in the paper is useless. It illustrates the variations in T and rainfall detected during the experiment compared to the historical data. However, after presenting the data in the text (rows 194-201), I would not waste a figure to reiterate the same data. For this reason, I would suggest dedicating one or two figures to the data presented in Table 3.

Regarding the issue of seasonal weather conditions, I have a more general comment. Climatic variability is one of the uncontrollable factors in open-field agronomic trials. Recent repeated meteorological anomalies make managing these aspects even more difficult. For the sake of providing significant and impactful results, it is crucial that field crop trials are conducted for 2-3 years. To underscore this, it's important to note that some scientific journals in the sector do not publish trials conducted for a single year. I am well aware of the reasons researchers neglect this aspect. However, the risks are illustrated by the authors of the MS.

c) paragraph 2.5 – Field experiment.

The authors report that ……. Application rates were determined based on agrochemical analyses conducted prior to maize sowing, reflecting to standard concentrations according to Slovak legislative norms.

That sounds good, but how many kg of N/P/K per hectare did you supply to the plants?

3. Results

a) paragraph 3.2Phytoavailability of Selected Soil Nutrients

It is unclear why data in Table 2 are included in the results. It seems more reasonable to me to include them in Table 1 Soil Parameters in M&M.

b) paragraph 3.3Effect of Foliar Application of Metal-Based Nanoparticles on Quantitative Parameters of Maize

The title of paragraph means nothing, just as I wouldn't know what "Agronomic effect" means.
In Field Crop research, we discuss Yield Components. In the case of corn, there are five: plants per acre, ears per plant, rows per ear, kernels per row, and kernel weight. Considering Table 3, it is clear that this is what we are talking about. I suggest rewording the text more clearly.

c) In the note below Table 3, the term variance is used incorrectly. The data in the table is simply mean ± standard deviation. Erase “variance”.

d) I can't evaluate the comments on Table 4 because it lacks statistics. Assuming the statistics do support these results, I'm surprised by the data on the concentration of Zn in maize grains. The concentration of Zn in the control group is lower than that of the foliar treatment with ZnO-NPs. I'm also seeing a similar trend with Si. Where did Si and Zn go? Can you provide an interpretation and explanation of this data?

4. Discussion

a) paragraph 4.1Assessment of Phytoavailability of Selected Nutrients from the Soil Environment for Maize Production Purposes

Another long text is not consistent with the experiment. The bioavailability of some soil elements has no relation to the experimental factors of this study, so these are not results.

Erase paragraph 4.1.

 b) paragraph 4.2 – Effects of Micronutrient-Based and Non-Essential Nano-Fertilizers on Maize Yield and Yield-forming Parameters

Again consider “Yield components” instead of Maize Yield and Yield-forming Parameters

c) paragraph 4.3/4.4

The absence of statistics in Table 4 prevents a critical evaluation of the data. Provide statistics.

Comments on the Quality of English Language

Minor editing of English language required

Author Response

Reviewer 1

# Comment 1. …While it's important not to stifle the progress of research, we must always remember that scientific research is not an end in itself. It's crucial to maintain a keen focus on reality, understanding the real-world implications of our findings, and the potential impact they can have on global agriculture and environmental sustainability.

# Answer 1. Thank you for your feedback and point of view. However, the focus of the article includes Au NPs selected as an “exotic” type of nanomaterial, as mentioned in paragraph L42-45 “…Some pioneering studies focus on applications to enhance crop production using plant-stimulators such as Si- or Ti-based materials [2] or “exotic” precious metal including gold nanoparticles (Au-NPs) [3]...”

Additionally, Au-NP is one of the most easily analytically detectable elements with potentially toxic translocation activity, which we have stated in the title “Enhancing Maize Yield and Quality with Metal-based Nanoparticles Without Translocation Risks: A Brief Field Study” and various sections of the manuscript, specifically in paragraph L 72 – 74 “Insoluble Au-NPs [14], relatively biologically stable SiO2 [15-17], or mechanically-chemically stable TiO2-NPs [18] hold a potential toxicological risk in comparison to more soluble micronutrient based ZnO-NPs [19].”

# Comment 2. rows 42-44: “Some pioneering studies focus on applications to enhance crop production using “exotic” materials, such as precious metals like Au, or Si- or Ti-based materials [1]”.

This statement is wrong. In the review cited on row 44 (Liu and Lal. 2015. STOTEN 514, 131-139. https://doi.org/10.1016/j.scitotenv.2015.01.104), the use of nAu is not mentioned. I suggest reformulating the sentence or, even better, supporting it with correct references.

# Answer 2. I agree, the text has been amended according to the reviewer’s recommendations. Please check # comment 1.

# Comment 3. rows 82-125: This section of the introduction is useless. Arguing on the physiological role of macronutrients in plant nutrition and the consequences on humans of a diet deficient/or excessively rich in the same elements is only helpful for filling lines of text.

For example: rows 100-101: Excessive doses (of Zn) are associated with nausea, vomiting or weakened immune system function. Delete this part, including the dozen or so literature quotes.

# Answer 3 We agree, in the relevant section of the manuscript, this was significantly reduced according to the reviewer’s recommendations.

# Comment 4. paragraph 2.3 - Experimental location description

Delete rows 172-178, not relevant. Insert data regarding soil texture (% sand, lime, clay) in Table 1.

# Answer 4. We agree, details about soil texture were added to Table 1 according to the reviewer’s suggestion. However, other parameters relate to the experimental locality description, such as soil conditions, geomorphology, geology, etc. we did not omit.

# Comment 5. paragraph 2.4 – Climatic seasonal variations during vegetation season 2021.

Apart from the SEM images, the only figure presented in the paper is useless. It illustrates the variations in T and rainfall detected during the experiment compared to the historical data. However, after presenting the data in the text (rows 194-201), I would not waste a figure to reiterate the same data. For this reason, I would suggest dedicating one or two figures to the data presented in Table 3.

# Answer 5. We agree, the duplicate text has been removed. However, we retained the comparison of temperature and precipitation trends with the long-term climate normal, which we also discuss in the text, e.g. L472-474 “During its key vegetative and reproductive phase of development, the synthesis and accumulation of sugars were relatively optimal in terms of temperatures based on monthly climatic normal (Figure 1a), but less favorable in terms of precipitation compared to long-term climatic normal (Figure 1b)”.

# Comment 6. Regarding the issue of seasonal weather conditions, I have a more general comment. Climatic variability is one of the uncontrollable factors in open-field agronomic trials. Recent repeated meteorological anomalies make managing these aspects even more difficult. For the sake of providing significant and impactful results, it is crucial that field crop trials are conducted for 2-3 years. To underscore this, it's important to note that some scientific journals in the sector do not publish trials conducted for a single year. I am well aware of the reasons researchers neglect this aspect. However, the risks are illustrated by the authors of the MS.

# Answer 6. Thank you for the recommendation. However, it is not possible to repeat the similar experiment on the same site and under the same conditions due to the crop rotation system. Additionally, there are frequent radical climate changes and meteorological anomalies at the experimental location. Therefore, we decided to present a one-year experimental trial. Our results, as a form of pioneering research, build upon our previous studies which have received significant citations, for example (Kolenčík et al. 2019, Kolenčík et al. 2020) or current study Kenyi et al. (2024).

# Comment 7. paragraph 2.5 – Field experiment. The authors report that ……. Application rates were determined based on agrochemical analyses conducted prior to maize sowing, reflecting to standard concentrations according to Slovak legislative norms. That sounds good, but how many kg of N/P/K per hectare did you supply to the plants?

# Answer 7. The information has been added, L205-206 “Informácia bola doplnená The NPK 15-15-15 DUSLOFERT fertilizer (Duslo, a. s., Šaľa, Slovak Republic) was dosed at 450 kg.ha−1.”

# Comment 8. paragraph 3.2 – Phytoavailability of Selected Soil Nutrients

It is unclear why data in Table 2 are included in the results. It seems more reasonable to me to include them in Table 1 Soil Parameters in M&M.

# Answer 8. We disagree, please check # Answer 12.

# Comment 9. paragraph 3.3 – Effect of Foliar Application of Metal-Based Nanoparticles on Quantitative Parameters of Maize

The title of paragraph means nothing, just as I wouldn't know what "Agronomic effect" means.
In Field Crop research, we discuss Yield Components. In the case of corn, there are five: plants per acre, ears per plant, rows per ear, kernels per row, and kernel weight. Considering Table 3, it is clear that this is what we are talking about. I suggest rewording the text more clearly.

# Answer 9. We agree, terminological inconsistencies have been corrected according to the reviewer's suggestions in several parts of the manuscript.

# Comment 10. In the note below Table 3, the term variance is used incorrectly. The data in the table is simply mean ± standard deviation. Erase “variance”.

# Answer 10. The term “variance” was removed as per the reviewer’s recommendation.

# Comment 11. I can’t evaluate the comments on Table 4 because it lacks statistics. Assuming the statistics do support these results, I’m surprised by the data on the concentration of Zn in maize grains. The concentration of Zn in the control group is lower than that of the foliar treatment with ZnO-NPs. I'm also seeing a similar trend with Si. Where did Si and Zn go? Can you provide an interpretation and explanation of this data?

# Answer 11. We agree, the statistics of mineral nutrients and the potential hazardous tranlocation of NPs or their residues have been added to Table 4.

Information about Si-NPs and Zn-NPs “destiny” was added into manuscript L545-547 “After foliar application, ZnO-NPs on maize may transform and distribute into zinc-soluble form, similarly like in sunflower leaves [79], potentially restricting their transfer into maize grains”, L 569-573 “Most probably, the phytoavailable Si was metabolized and used in formation of plant tissues, such as leaves during the formation of phytoliths, trichomes, ear, or supporting the formation of phytoplasmic bodies or mechanical structures in plants, where Si is preferentially accumulated in higher concentrations [70]”.

# Comment 12. paragraph 4.1 – Assessment of Phytoavailability of Selected Nutrients from the Soil Environment for Maize Production Purposes

Another long text is not consistent with the experiment. The bioavailability of some soil elements has no relation to the experimental factors of this study, so these are not results.

Erase paragraph 4.1.

# Answer 12. We disagree. The presence and phytoavailability of mineral nutrients in soil, i.e., soil chemistry, directly correlates with the quality of the monitored final products. These findings are highlighted in several sections of the manuscript, e.g. L352-355 “Soil chemical composition, particularly the phytoavailable fraction, influences mineral nutrients uptake, such as in maize [51], thereby affecting nutritional value and final grain-quality with human impact [52].”

# Comment 13. paragraph 4.2 – Effects of Micronutrient-Based and Non-Essential Nano-Fertilizers on Maize Yield and Yield-forming Parameters

Again consider “Yield components” instead of Maize Yield and Yield-forming Parameters

# Answer 13. We concur. In the manuscript, the terms were adjusted according to the reviewer’s recommendations, similar to Chapter 3.3.

# Comment 14. paragraph 4.3/4.4

The absence of statistics in Table 4 prevents a critical evaluation of the data. Provide statistics.

# Answer 14. We agree; statistical analysis has been incorporated.

Reviewer 2 Report

Comments and Suggestions for Authors

Comments on the Quality of English Language

There are many small paragraphs in the text (only 2-3 lines). I think it is very unfriendly to readers and unable to help readers understand the whole text well. It's best for authors to rearrange them.

Author Response

Reviewer 2

# Comment 15. please reorganize key words.

# Answer 15. We agree; key terms have been modified as per the reviewer’s suggestions L36-37 “Keywords: spray application; maize; yield; grain quality; nanofertilizers; nanoparticles; gold; biosilica; zinc oxide; titanium dioxide”.

# Comment 16. Introduction: In this section, the author excessively explains the effects of trace elements on the human body, however, we need to note that the theme of the article is the impact of nanomaterials on crop yield and quality. Please adjust some content to ensure it matches the Topic. This is very important!

# Answer 16. Since we discuss the profile of mineral nutrients in maize, including the potential toxic transport of gold, in the abstract, objectives, results, and discussion sections, we have left their impact on humans in the introduction section. However, we have significantly reduced this portion based on the reviewer’s recommendations.

# Comment 17. L148: Please confirm if “ζ-potential” here is correct.

# Answer 17. Yes, we are referring to the zeta potential.

# Comment 18. Table 1  “pH H2O” should be changed to “pH H2O”.

# Answer 18. We agree; it has been adjusted according to the reviewer’s recommendations.

# Comment 19. The discussion should be more targeted and deliberative.

# Answer 19. We agree; revisions have been made throughout the manuscript based on the reviewers’ recommendations. For further details, please check Reviewer 1, Comment #11, Comment #12.

# Comment 20. There are many small paragraphs in the text (only 2-3 lines). I think it is very unfriendly to readers and unable to help readers understand the whole text well. It's best for authors to rearrange them.

# Answer 20. We agree; it has been adjusted according to the reviewer’s recommendations.

Reviewer 3 Report

Comments and Suggestions for Authors

This article investigates the effects of zinc oxide, Au NP bioSi, and TiO2 NPs on maize growth, nutritional quality, and yield under field conditions, and evaluates their potential risks. The article closely follows current research hotspots and has achieved some valuable results. I suggest publishing the article after revision.

1, The scale of the electron microscope in Figure 2 is unclear, and the size of the nanoparticles cannot be seen.

2, There are indexing issues in the writing of the article, such as pages 291, 296, 297.

3,  The unit should be in international standard units. For example, hectares are hm2, not ha.

4, The latest relevant references should be fully discussed.  DOI10.3390/nano12234160; DOI10.3390/nano12234219

Author Response

# Comment 21. The scale of the electron microscope in Figure 2 is unclear, and the size of the nanoparticles cannot be seen.

# Answer 21. I agree, selected sections of SEM and EDS images were highlighted.

# Comment 22. 2, There are indexing issues in the writing of the article, such as pages 291, 296, 297.

# Answer 22. The indexes were corrected.

# Comment 23. 3, The unit should be in international standard units. For example, hectares are hm2, not ha.

# Answer 23. We do not plan to modify the units as they comply with the MDPI standard published in the following manuscripts (Kenyi et al. 2024, Zagyi et al. 2024).

# Comment 24. The latest relevant references should be fully discussed.  DOI10.3390/nano12234160; DOI10.3390/nano12234219

# Answer 24. Both publications have been incorporated into the manuscript and enriched the introduction and discussion sections.

Round 2

Reviewer 2 Report

Comments and Suggestions for Authors

Line 137:“a ζ-potential potential ” is right?

Table 1: It is also the "pH H2O in the new version"

Figure 2: The layout of this figure is very messy in my side. Please check again.

Comments on the Quality of English Language

It is Ok.

Author Response

Reviewer 1

# comment 1

Line 137:“a ζ-potential potential ” is right?

# Answer 1. We agree, the text has been revised according to the reviewer's recommendations.

# comment 2

Table 1: It is also the "pH H2O in the new version"

# Answer 2. We agree, the text has been revised according to the reviewer's recommendations.

# comment 3

Figure 2: The layout of this figure is very messy in my side. Please check again.

# Answer 3. We agree, the text has been revised according to the reviewer's recommendations.

In the introduction and conclusion chapters, abbreviations, English language review in the text were also modified according to the reviewer's recommendations.